# System Identification and Mathematical Modeling of A Piezoelectric Actuator through A Practical Three-Stage Mechanism

**DOI:** 10.3390/mi14010088

**Published:** 2022-12-29

**Authors:** Dror A. Levy, Amir Shapiro

**Affiliations:** Department of Mechanical Engineering, Ben-Gurion University of the Negev, Beer-Sheva 84105, Israel

**Keywords:** amplification mechanism, analytical models, hysteresis, modelling, piezoelectric actuators, system identification

## Abstract

Piezoelectric elements (PEMs) are used in a variety of applications. In this paper, we developed a full analytical model and a simple system identification (SI) method of a piezoelectric actuator, which includes piezostack elements and a three-stage amplification mechanism. The model was derived separately for each unit of the system. Next, the units were combined, while taking into account their coupling. The hysteresis phenomenon, which is significant in piezoelectric materials, is described extensively. The theoretical model was verified in a laboratory setup. This setup includes a piezoelectric actuator, measuring devices and an acquisition system. The measured results were compared to the theoretical results. Some of the most well-known forms of system identification are shown briefly, while a new and simple algorithm is described systematically and verified by the model. The main advantage of this work is to provide a solid background and domain knowledge of modelling and system identification methods for further investigations in the field of piezoelectric actuators. Due to their simplicity, both the model and the system identification method can be easily modified in order to be applied to other PEMs or other amplification mechanism methods. The main novelty of this work lies in applying a simple system identification algorithm while using the system-level approach for piezoelectric actuators. Lastly, this review work is concluded and some recommendations for researchers working in this area are presented.

## 1. Introduction

The review provided in this paper is intended to give insights into system identification approaches as an alternative way to obtain mathematical models for a piezoelectric actuator through a practical three-stage mechanism based on experimental data. Such approaches differ from the conventional modeling approaches through physics. With system identification approaches, the richness of experimental data can be exploited to obtain the dynamics and nonlinear characteristics of these piezoelectric actuators. The models obtained through such approaches can be used to validate mathematical models and predict the effect of any design changes.

The piezoelectric effect describes the electric field created when a piezoelectric element is mechanically loaded. On the other hand, the reverse piezoelectric effect describes the deformation of the piezoelectric element when an electric voltage is applied to its electrodes [1]. For mounting piezoelectric elements on any mechanical environment, let us refer to the system as a piezoelectric actuator (PEA).

PEAs are very common in a wide variety of applications, especially in science, industry, and medicine. PEAs can be used as actuators [2], such as in piezoelectric motors [3], piezoelectric ultrasonic motors [4], inertia drives [5], cantilever beams [6], inchworm actuator [7], industrial robotics [8], sensors [9], micro-positioning [10], micro-cantilever probes [11], nano-positioners [12], medical engineering [13], fuel injectors [14], aeronautic applications such as helicopters [15] and space applications [16], ultrasound equipment [17], vibration systems [18,19], optical communications [20], the control of pneumatic actuators [21], transformers [22,23], and more.

The widespread use of PEAs is due to their many benefits which are reflected in the following parameters: high efficiency, a short response time, high resolution, the ability to produce large forces, insensitivity to ambient temperature, no production of a magnetic field, and the absence of bearings, cogwheels, or other moving parts that can be eroded. In addition, PEAs are attractive because their electrical behavior is similar to that of a capacitor, while there is no power consumption in steady-state conditions.

In the research described in this paper we used a piezoelectric actuator which consists of three PEMs which are mounted through a practical mechanism. The amplification mechanism consists of three amplification stages, mounted on one another. Each stage acts like a lever with nonlinear properties, because of the non-constant position of the contact points between the levers, as shown in Figure 1.

These levers convert linear movements of 50 μm to angular movements of 40 degrees. The linear displacement of three parallel piezostacks produces the rotational movement of lever #1 and lever #2 and finally causes the rotation of the output axis.

In a previous piece of work [24], a detailed model and mathematical analysis of the integration of PEMs in the amplification mechanism and the coupling between them as well as between the applied load were presented. In addition, the model deals with several nonlinear phenomena that can be found in such systems. One of the more common phenomena in PEMs is the hysteresis phenomenon, which was studied in depth in a previous article by researchers [24].

The system identification method is a mathematical technique based on observed information about the inputs and outputs of the system. This method was first introduced by LoftiZadeh in the 1950s [25]. In addition, since then it has been used to describe and identify parameters of complex systems, such as aircraft [26,27], underwater vehicles [28], and cardiovascular system [29].

The main aim of the model approach is to develop a mathematical description of a complex system based on experimental data, in order to use it in the fields of simulations, prediction, and control [30].

There are two ways of using the system identification: the black box approach in which system identification is carried out by fitting the system’s input–output response with some experimental data, or the grey box approach [31], which is carried out by first developing the structure of models mathematically, then the required parameters in the models are identified. In both approaches, the overall models are validated against the experimental data. The typical steps involved in system identification are shown in Figure 2.

In this paper, the most prominent strategies relevant to system identification of a piezoelectric actuator through a practical three-stage mechanism were overviewed.

The ultimate objective, in the development of PEMs, is to utilize them for the different tasks mentioned earlier. Therefore, the first step is the modeling of the PEMs’ behavior. This task is particularly challenging due to nonlinearity, time variance, and the hysteresis phenomena of the PEMs [24]. These phenomena have not been fully uncovered yet.

However, in this review, we focused on system identification approaches in order to model the PEMs without actually going through the details of those phenomena, besides the hysteresis that was described in detail in a previous article [24].

In order to develop a PEA model, two complementary strategies are proposed. The first strategy is called “bottom-up”, in which we focused on the fundamental approaches to explain the behavior of the PEAs, while using the studied physics of the PEAs. On the other hand, the second strategy is called a “top-down” approach, where the construction and development of a real PEA precedes the theoretical development.

System identification is the first step in the design of a complete mathematical model of any system, when combining the initial analytical information as well as the system measurement data together, which could be in the time domain or in the frequency domain.

This approach has several advantages: the effect of each of the parameters on the system can be tested, changes in the system can be taken into account, and control loops can be more easily designed. The structure of the mathematical models obtained is not unique, which gives freedom to the engineers/researchers to select the models that suit the objectives of the modeling.

The PEMs and their behavior have been elucidated in a general way, and now the objective is to present the system identification in detail. It is obvious that the dynamic modeling of PEMs always considers the mechanism linkages, which connect the actuator and the load in the last amplification stage, and nonlinear phenomena such as saturation and hysteresis [24]. Consequently, it results in different equations and realizations of the dynamics.

The rest of this paper is organized as follows. Section 2 contains a discussion on system identification methods, which then leads to the main review that presents different strategies in system identification adapted for modeling the PEMs.

In Section 3, we present and analyze the combined system model including a description of the PEMs, the hysteresis phenomenon in piezoelectric materials, and the electromechanical coupling. (This section is a brief conclusion of previous work [24]). In Section 4, the system identification process is described in detail and applied. Each step is described separately, when the products of each stage are the source of the data for the next stage, until the complete process is obtained, from measuring the data to obtaining the system identification of the whole system. Finally, our concluding remarks and some recommendations of proposed strategies for future works and research in the field of the system identification of a piezoelectric actuator through a practical three-stage mechanism are presented in Section 5.

## 2. System Identification Methods

There are a large number of system identification methods [32,33]. In this article, four of the most common detection methods have been presented. These are the least-squares complex exponential (LSCE) method, the Ibrahim time domain (ITD) method, the frequency domain direct parameter identification (FDPI) method, and the Least-Squares complex frequency-domain (LSCF) method. It is obvious that each of the methods has its advantages and disadvantages, in terms of the simplicity and efficiency of the calculation, the accuracy of the results, etc.

The least-squares complex exponential (LSCE) method is considered as an “industry-standard” time-domain estimation technique [34,35,36,37,38]. The LSCE method is therefore a single-input multi-output (SIMO) technique, handling several impulse response functions (IRFs) at the same time obtained by exciting a structure at one single point and estimating the responses at several locations. The LSCE estimates the system poles in the time space by using several IRFs obtained from frequency response functions (FRFs) by an inverse Fourier transform (IFFT).

The Ibrahim time domain (ITD) method [39,40] was developed in order to overcome processing issues encountered using frequency domain methods. Some of the ITD method’s advantages are the lack of a necessity to use Fourier transform. In addition, the input excitation does not need to be measured, which makes the ITD method attractive for operational modal analysis applications. However, care must be taken when acquiring the free responses. These latter aspects have to be measured after the initial exciting force is removed.

The frequency-domain direct parameter identification (FDPI) technique has usually been used to analyze data from highly damped structures [41,42,43]. A famous advantage of FDPI is that real normal modes can be identified directly by solving the undammed eigenvalue problem using just the mass modified stiffness matrix. This is clearly an advantage over other modal identification methods, where real normal modes are usually derived from complex ones by forcing certain suppositions.

The least-squares complex frequency-domain method known as LSCF or as PolyMax can be considered as a frequency domain implementation of the LSCE estimator [44,45,46]. The PolyMax technique shows several advantages such as the use of frequency-dependent weighting functions. Moreover, the most important advantage of the PolyMax estimator is maybe the fact that it produces “fast-stabilizing” stabilization charts. The stability property of the system poles is an interesting argument which offers the possibility to automatically remove unstable poles from the stabilization diagram. This can be considered as one of the most important advantage the PolyMax technique over other modal analysis methods.

Following these works, and in order to better facilitate the system identification process of PEMs, we developed a new and simpler algorithm.

The algorithm is based on a system-oriented approach which has been described in detail in previous work. The algorithms consist of three simple stages that finally provide us with the identification of the whole actuator through a practical three-stage mechanism.

In general, this algorithm uses a *MATLAB* function: *invfreqs* [47], which is based on the LSCF method, with some simple mathematical calculations.

By default, invfreqs uses an equation error method to identify the best model from the data. This finds b and a in Equation (1).
(1)$$\underset{b,a}{\min}\overset{n}{\underset{k=1}{\sum }}wt\left(k\right){\left|h\left(k\right)A\left(w\left(k\right)\right)-B\left(w\left(k\right)\right)\right|}^{2}$$

A system of linear equations is created and they are solved with the MATLAB^®^\operator. Here A(w(k)) and B(w(k)) are the Fourier transforms of the polynomials *a* and *b*, respectively, at the frequency w(k), and n is the number of frequency points (the length of *h* and *w*).

The superior (“output-error”) algorithm uses the damped Gauss–Newton method for iterative search [48], with the output of the first algorithm as the initial estimate. This solves the direct problem of minimizing the weighted sum of the squared error between the actual and the desired frequency response points, as shown in Equation (2).
(2)$$\underset{b,a}{\min}\overset{n}{\underset{k=1}{\sum }}wt\left(k\right){\left|h\left(k\right)-\frac{B\left(w\left(k\right)\right)}{A\left(w\left(k\right)\right)}\right|}^{2}$$

Applying system identification involves a set of problems that must be taken into account: mainly providing a suitable data set, choosing the most suitable identification method, and specifying the model structure.

The system identification algorithm is fully described in Section 4.

## 3. The Combined System Model

The combine system model, presented in detail in a previous piece of work [24], refers to the integrated system as one unit. The input to the system is the voltage applied on the electrodes of the PEMs, and the output of the system is the angular movement at the end of the third amplification stage of the amplification mechanism. This movement is also affected by the load as well as the initial conditions and the displacement and the PEM as can be seen in Figure 3.

### 3.1. Piezoelectric Elements (PEMs)

As mentioned earlier, a mechanical load applied to the PEM creates an electric field on the electrodes of the PEMs. The direction of the electric field depends on the polarity of the PEMs.

In general, PEMs are represented in terms of three-directional axes and three directions of rotation [49]. The indexes 1, 2, and 3 represent the three-dimensional axis and 4, 5, and 6 represents the rotary axis, as seen in Figure 4.

In our system we use one of the well-known and widely used versions of PEMs, the piezostack [51]. In this case, the directions of the mechanical load and the electric field are along one axis.

A simple piezostack model was described in detail in a previous piece of work [24]. The outcome equations of this model can be separated into two parts, electrical and mechanical.

The electrical part is:(3)$$Vact=\frac{Vin-R\cdot Cm\cdot d_{33}\cdot v}{R\cdot C\cdot S+1}$$

The mechanical part is:(4)$$Fact=Vact\cdot Cm\cdot d_{33}-Fext=\left(m\cdot S^{2}+Kd\cdot S+Cm\right)\cdot x$$
which can be combined into one equation:(5)$$\left(\frac{Vin-R\cdot Cm\cdot d_{33}\cdot S\cdot x}{R\cdot C\cdot S+1}\right)-Fext=\left(m\cdot S^{2}+Kd\cdot S+Cm\right)\cdot x$$

The piezoelectric stack was created in Simulink, as shown in the following block diagram (Figure 5) [24]. The variables and parameters in this model are described in Table 1 and Table 2.

### 3.2. Hysteresis in Piezoelectric Materials

A significant drawback of PEMs that must be considered in any design is hysteresis, that can significantly degrade the performance of the PEAs, which leads, at best, to a reduction in the motion accuracy and, at worst, to destabilization of the control system [52].

In a previous piece of work, we performed an extensive survey of the hysteresis phenomenon of piezoelectric actuators. In the work, several approaches were presented to describe this phenomenon, with the most famous of them being the Preisach model of hysteresis [52,53,54,55,56], the generalized Maxwell slip (GMS) model of hysteresis [57], and the Prandtl–Ishlinskii operator [58,59,60,61].

In Figure 6, the classical hysteresis curve is displayed [62]. Assuming that the relation between the driving voltage and the PEA displacement is linear, the hysteresis curve has been reduced to a linear function (the dashed red line). In fact, when the driving voltage increases, the curve which describes the voltage-displacement relation is below the linear relation, and when the driving voltage decreases, it is above the linear relation.

Another method uses one measured curve and defines a mathematical process in order to evaluate any other possible curve [63]. In order to better facilitate the modeling of the hysteresis phenomenon, we developed a new and simpler algorithm, which determines one normalized basic curve and calculates the numerical relation to any other possible curve. We make this calculation by measuring the input signal and defining its direction as a function of time, without knowing its minimum and maximum values.

In general, this algorithm considers the tendency of the voltage applied on the piezostack. Each of these curves is described by a set of two equations, one for the positive derivative and the other for the negative derivative. The algorithm is described in detail in a previous piece of work [24].

To easily describe the hysteresis phenomenon in PEMs, a set of equations developed in a previous piece of work (6,7) can be used [24]. The equation constants are logical laws about the values of *Vmax* and *Vmin* and they depend on the tendency of the input voltage. The parameters of the equation are shown in Table 3.
(6)$$\begin{array}{cc} Hist_{Up}\left(v_{in},v_{min}\right) & \\ & =P_{up_{1}}\cdot {\left(v_{in}-v_{min}\right)}^{3}+P_{up_{2}}\cdot {\left(v_{in}-v_{min}\right)}^{2}+P_{up_{3}}\cdot \left(v_{in}-v_{min}\right)+P_{up_{4}}+B_{up_{1}}\cdot {\left(v_{min}\right)}^{2}\\ & +B_{up_{2}}\cdot \left(v_{min}\right)+B_{up_{3}} \end{array}$$
(7)$$\begin{array}{cc} Hist_{Dn}\left(v_{in},v_{max}\right) & \\ & =P_{dn_{1}}\cdot {\left(v_{in}-v_{max}\right)}^{4}+P_{dn_{2}}\cdot {\left(v_{in}-v_{max}\right)}^{3}+P_{dn_{3}}\cdot {\left(v_{in}-v_{max}\right)}^{2}+P_{dn_{4}}\cdot \left(v_{in}-v_{max}\right)\\ & +P_{dn_{5}}-B_{dn_{1}}\cdot {\left(v_{max}\right)}^{2}+B_{dn_{2}}\cdot \left(v_{max}\right)+B_{dn_{3}} \end{array}$$

### 3.3. Amplification Mechanism

There are various methods of the amplification mechanism; the most well-known and useful method is the lever. In the case discussed in this article, the amplification system consists of three levers placed on top of each other. Each lever acts like a nonlinear lever because of the non-constant position of the touch position between the levers. The first lever converts the PEMs’ linear displacement from 50 µm to an angular movement of 1.5 degrees. The second lever converts the results of the first lever from 1.5 degrees to 20 degrees. In the same way, the third lever converts the results of the second lever from 20 degrees to 40 degrees. The full range amplification is from a linear displacement of 50 μm to an angular movement of 40 degrees. Another parameter of the amplification mechanism is the pre-loading, which is used in order to determine the “zero-point” offset, as shown in Figure 7.

The ship between the PEM displacement (*x*) and the angular angle in the third levers ($\alpha_{out}$) is described in following equations:(8)$$L_{1}=\sqrt{{\left(X_{13}-\frac{X_{12}\cdot Rb_{1}-Y_{12}\cdot x}{\sqrt{R{b_{1}}^{2}+x^{2}}}\right)}^{2}+{\left(Y_{13}-\frac{Y_{12}\cdot h_{0}+X_{12}\cdot x}{\sqrt{R{b_{1}}^{2}+x^{2}}}\right)}^{2}-{\left(Rb_{2}\right)}^{2}}$$
(9)$$\alpha_{34}=\sin^{-1}\left(\frac{L_{1}\cdot \left(Y_{13}-\frac{Y_{12}\cdot h_{0}+X_{12}\cdot x}{\sqrt{R{b_{1}}^{2}+x^{2}}}\right)-\left(Rb_{2}\right)\cdot \left(X_{13}-\frac{X_{12}\cdot Rb_{1}-Y_{12}\cdot x}{\sqrt{R{b_{1}}^{2}+x^{2}}}\right)}{{\left(X_{13}-\frac{X_{12}\cdot Rb_{1}-Y_{12}\cdot x}{\sqrt{R{b_{1}}^{2}+x^{2}}}\right)}^{2}+{\left(Y_{13}-\frac{Y_{12}\cdot h_{0}+X_{12}\cdot x}{\sqrt{R{b_{1}}^{2}+x^{2}}}\right)}^{2}}\right)+\alpha_{0}$$
(10)$$L_{2}=\sqrt{{\left(X_{35}-R_{34}\cdot \sin\left(\alpha_{34}\right)\right)}^{2}+{\left(Y_{35}-R_{34}\cdot \cos\left(\alpha_{34}\right)\right)}^{2}-{\left(Rb_{3}\right)}^{2}}$$
(11)$$\sin\left(\alpha_{out}\right)=\sqrt{\frac{-\left(Rb_{3}\right)\cdot \left(X_{35}-R_{34}\cdot \sin\left(\alpha_{34}\right)\right)+L_{2}\cdot \left(Y_{35}-R_{34}\cdot \cos\left(\alpha_{34}\right)\right)}{{\left(X_{35}-R_{34}\cdot \sin\left(\alpha_{34}\right)\right)}^{2}+{\left(Y_{35}-R_{34}\cdot \cos\left(\alpha_{34}\right)\right)}^{2}}}$$

The inputs to the equation are x and $\alpha_{0}$, which are the PEM displacement [mm] and the initial angular position [deg], respectively.

The results of the calculations based on these equations are shown in Table 4.

The parameters of the amplification mechanism equations are shown in Table 5.

The reflection of the external load on the PEMs can be described by Equation (12).
(12)$$Ext\_Tor\_Reflaction=Ext\_Tor\cdot \frac{R_{12}\cdot R_{34}}{Rb_{1}\cdot L_{1}\cdot L_{2}}$$

A detailed explanation can be found in a previous article by the authors [24].

### 3.4. Electro-Mechanical Coupling

The coupling between the applied stress and the mechanical output of the PEA has been described in detail in a previous article by researchers [24]. In general, the voltage applied to the PEM electrodes creates an electric field that causes polarization on the PEM electrodes, as for example in capacitors. This field, according to the inverse piezoelectric effect, causes displacement in the PEM. However, on the other hand, according to the piezoelectric effect, the voltage on the electrodes of the PEM decreases due to the displacement and load. Moreover, PEM displacement is also affected by the friction (Kd), stiffness (Cm), and piezoelectric constant $\left(d_{33}\right)$ of the specific PEM.

In addition, there is the mechanical amplification which converts, through the levers, the displacement of the PEM (*x*) from linear values of 50 um to an angular value ($\alpha_{out}$) of 40 degrees, in the third stage. As shown in Figure 8 [24].

### 3.5. Combined System Model

Analyzing each part of the system separately leads us to building the whole system. The parts of the system are the PEMs, the calculation of the angular movement caused by the displacement, the levers, and the reflection of the load on the PEMs according to the levers’ position, as shown in Figure 9.

## 4. System Identification Process Description

### 4.1. System Measurements

In order to obtain the database in which the process will be carried out, a series of measurements was completed. In the first stage, chirp measurements were performed at different input voltages and at different loads. The test frequency of the chirp ranges from 0.5 Hz to 20 Hz. This frequency range was chosen because at higher frequencies, the attenuation of the system is very large. Measurements above this frequency are not comparable.

The measurements are of the output angle of the actuator at the third-degree end of the mechanical amplification system and the current measured on the actuator terminals, depending on the input voltage, as we can see in Figure 10 and Figure 11. These measurements are the database for the entire system identification algorithm and the systemic model, when the input data to the algorithm and the system model are the input voltage and the load and the output is the angle and the current, which are ultimately compared to the measured data.

After deciphering the measurements and graphically showing the dependence of the output angle and current on the input voltage and load, the results could be processed and the system characterization could be obtained.

### 4.2. System Characterization

#### 4.2.1. Evaluation of Transfer Function

In the first step, a transfer function was performed, which will suit each of the system’s input voltages. The estimation was performed by activating the fast Fourier transform function on the input voltage values and the angle and current feedbacks. At the end of this, the results of the measurements in the frequency plane were obtained, as we can see in Figure 12 and Figure 13.

The transfer function was then re-evaluated using a Matlab function called Invfreqs. This function completes calculations using the least-square analog filter corresponding to the input and output signals, as fully described in Section 2.

#### 4.2.2. Position Feedback Transfer Function Adjustment

Following the calculation of a uniform and normalized transfer function, which is suitable for the different voltage levels and the different loads, calibration of the function for the various input voltages was performed. That is, for each input voltage the gain in which the uniform function was multiplied is determined. The amplification is determined by the following equation:
(13)$$Gain=0.0047\cdot Vin^{2}-0.724\cdot Vin$$

In order to test the suitability of the overall function of the amplifier for each input voltage, a comparison was made between the position feedback evaluation obtained from the operation of the general estimation function and the amplifier dependent on the input voltage of the input voltage values and the angle measurement at the mechanical amplifier output. These results are shown in Figure 14 and Figure 15.

#### 4.2.3. Current Feedback Transfer Function Adjustment

Unlike the approximation for position feedback, it turns out that the approximation for current feedback is much simpler. Generally, in piezoelectric elements the current is treated as in a series RLC circuit [64]. In our case, according to the manufacturer’s settings of the component in the datasheet, the resonance frequency is 50 kHz, the capacitance is 3.4 µF, and the frequency at which the input voltage frequency is less than 20 Hz, Therefore, the inductance component (L) can be ignored, and the piezoelectric element can be treated as a pure capacitive component.

Therefore, the transfer function in the Laplace domain of the current as a function of the input voltage is:
(14)$$\frac{I\left(s\right)}{V\left(s\right)}=\left(28.5\cdot 10^{-6}\right)\cdot S$$

Such as in the position case, a comparison of current feedback was made between the revaluation of the current feedback obtained from the operation of the general revaluation function on the input voltage values and the measurement of the current on the operator. These results are shown in Figure 16 and Figure 17.

### 4.3. Model Verification

In the first article [24], a complete and detailed system model of the system including a piezoelectric actuator and a three-degree mechanical amplification system was presented. The system description also included a full analytical analysis of the hysteresis phenomenon in the piezoelectric actuators. In this work, the measured input voltages were the input of the complete system model, and the results obtained from the model were compared to the measured results and the results obtained from the estimation function of the angle and current feedback. This comparison can be seen Figure 18 and Figure 19.

It can be seen that there Is a good match between the results of the evaluation functions of the position and current feedback and the results obtained from the full system model and the actual measurements. Despite this there are several small deviations between the model and the measured results, which probably result from measurement noise, static friction, touch, or the other nonlinear characteristics of the amplification mechanism that are beyond the scope of this article. Using a simple closed loop control method can overcome these gaps, where the input will be the desired angular output and the control loop will calculate the appropriate voltage input to achieve this output angle.

## 5. Conclusions

A comprehensive review on system identification and mathematical modeling of a piezoelectric multilayer stack actuator has been presented. The model was developed to provide a prediction of the output response and the optimal working conditions of the actuator. The model consists of some nonlinear elements. A detailed analytical calculation of the mechanical amplification mechanism and hysteresis phenomena was demonstrated. The obtained models were validated to predict and/or estimate the dynamics of the plants used, and they could achieve reasonable accuracies. The identification strategies that have been used were top-down grey box approaches.

Measurements of the actuator output and the piezoelectric stacks are reported for commercially available piezoelectric stack elements. The measured data were used in order to check and to tune some parameters in the model. The system identification algorithm and the model were shown to provide satisfactory predictions of the actuator’s overall performance.

This review has shown that system identification approaches have a promising potential to uncover many unanswered questions in generating valid mathematical models of piezoelectric actuators. Better outcomes of system identification and modeling complex systems such as piezoelectric actuators can be achieved with detailed analyses completed before the start of the identification which lead to appropriate model selection. Such analyses will be critical to determine the optimized model structure and model order where a priori knowledge and engineering insights are combined with the formal properties of the models. In addition, it is imperative to have clear objectives for the system identification to be performed; for example, system identification for design decisions, simulations, or validation, and/or for flight control design. Therefore, specific experimental data, efficient model selection, and structure as well as an identification approach can be determined accordingly.

## Figures and Tables

**Figure 1 micromachines-14-00088-f001:**
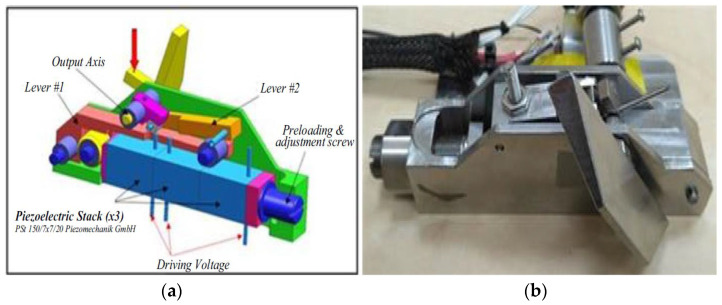
Three-stage lever scheme (**a**) and picture (**b**) [21].

**Figure 2 micromachines-14-00088-f002:**
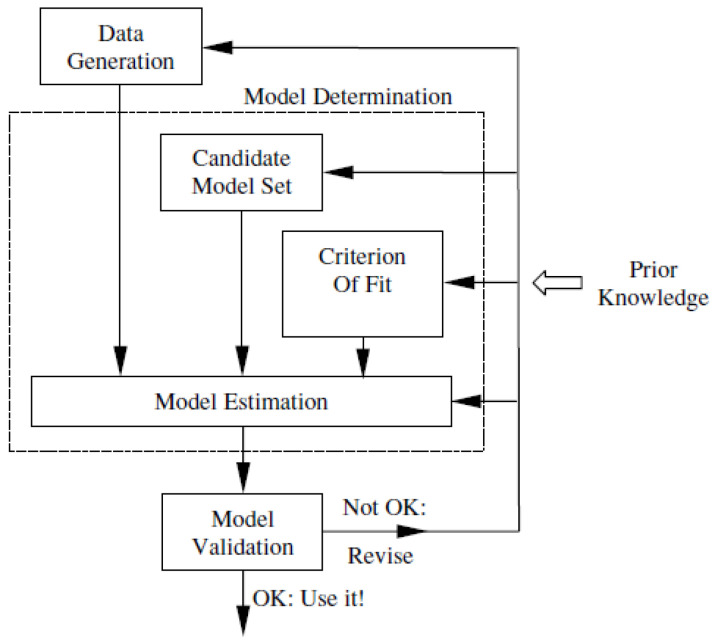
The system identification loop [26].

**Figure 3 micromachines-14-00088-f003:**
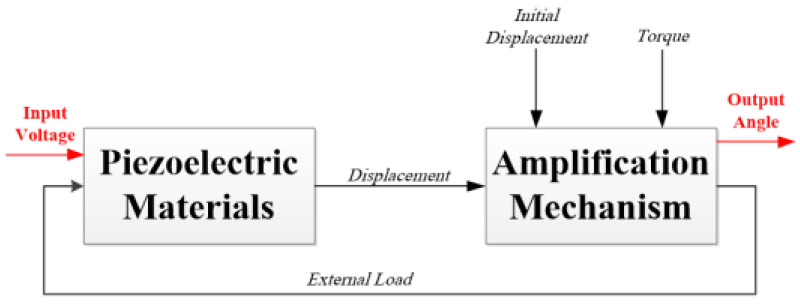
The combined system scheme [24].

**Figure 4 micromachines-14-00088-f004:**
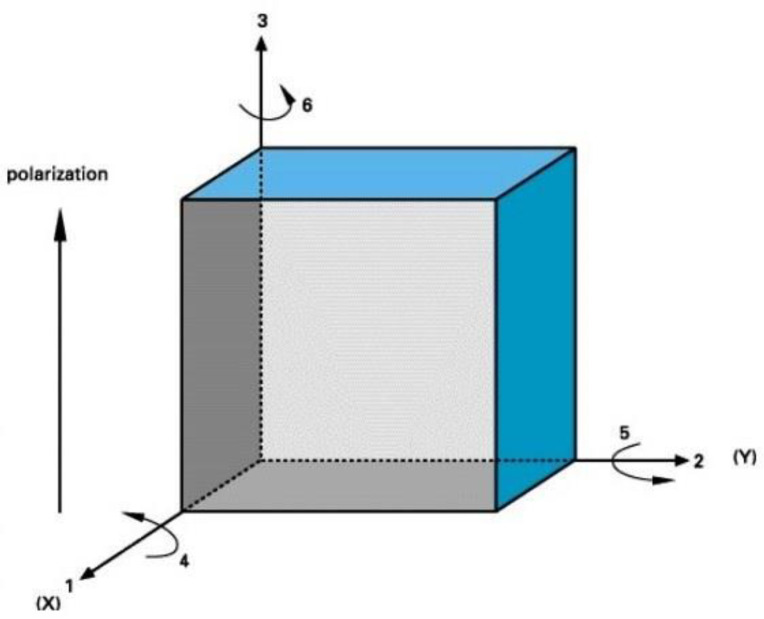
Poled piezoelectric ceramic [50].

**Figure 5 micromachines-14-00088-f005:**
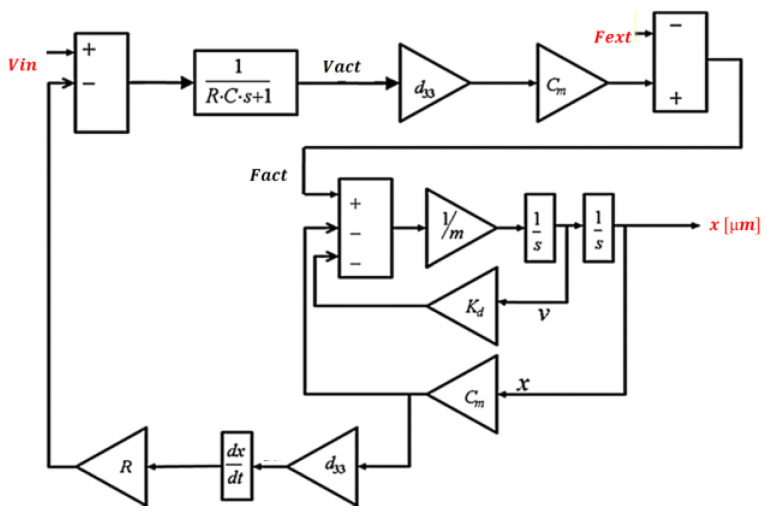
Simulink piezostack model.

**Figure 6 micromachines-14-00088-f006:**
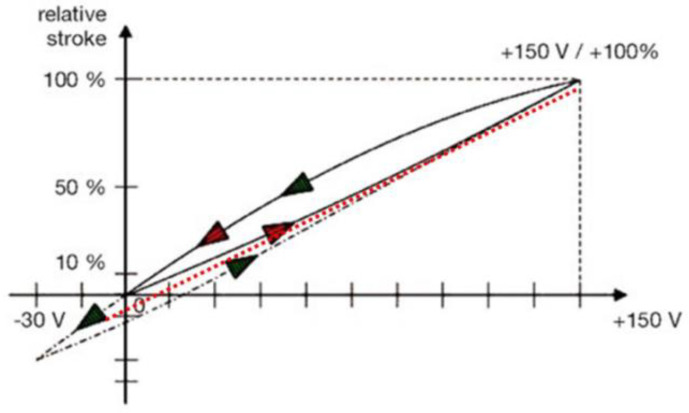
Classical hysteresis curve of PEMs [62].

**Figure 7 micromachines-14-00088-f007:**
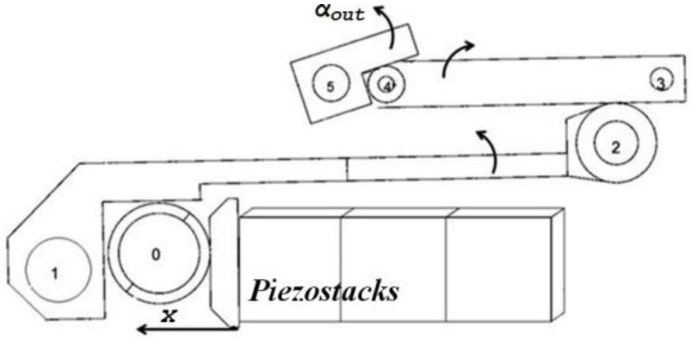
The mechanical amplification [24].

**Figure 8 micromachines-14-00088-f008:**
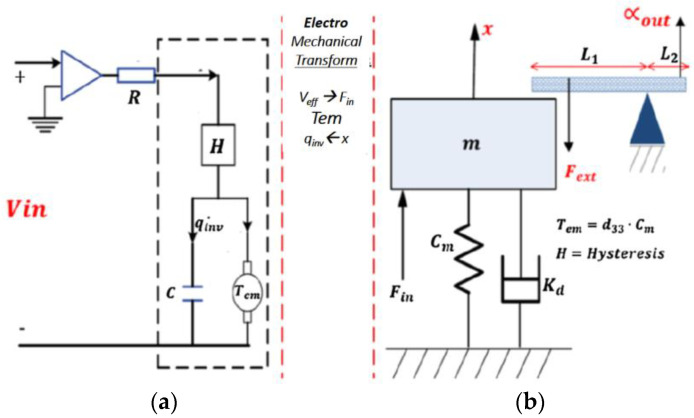
General model of piezoelectric positioning stage: (**a**) electrical scheme and (**b**) mechanical scheme [24].

**Figure 9 micromachines-14-00088-f009:**
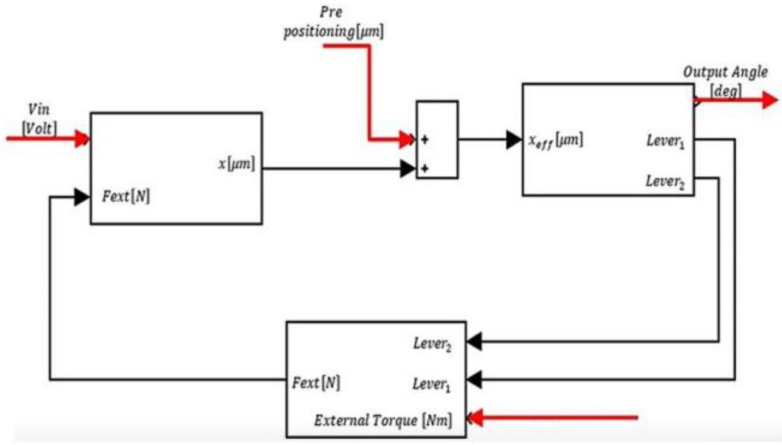
The combined system model [24].

**Figure 10 micromachines-14-00088-f010:**
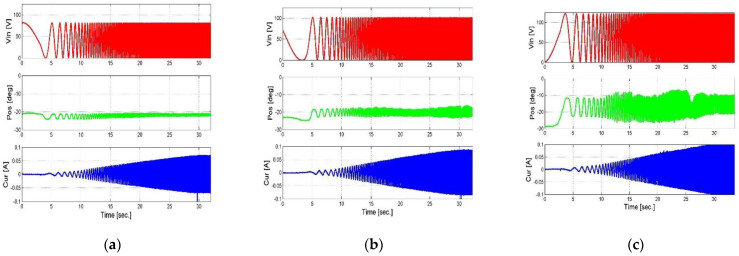
Measurement of output angle and current as results of the chirp input signal when the first load is applied. (**a**) Vin = 80 V, (**b**) Vin = 100 V, and (**c**) Vin = 120 V.

**Figure 11 micromachines-14-00088-f011:**
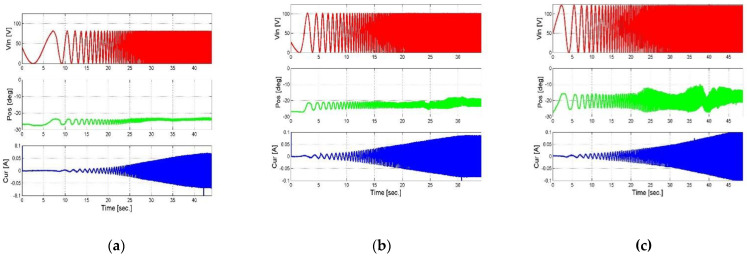
Measurement of output angle and current as results of the chirp input signal when the second load is applied. (**a**) Vin = 80 V, (**b**) Vin = 100 V, (**c**) Vin = 120 V.

**Figure 12 micromachines-14-00088-f012:**
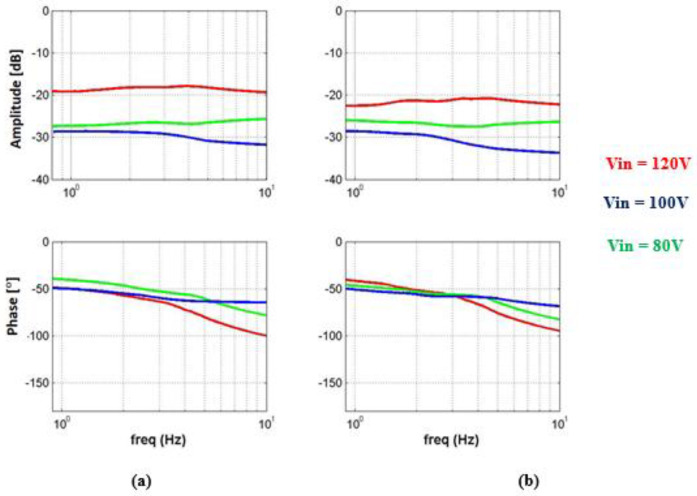
Bode plot of the output angle as a function of input voltage amplitude and load: (**a**) First load and (**b**) second load.

**Figure 13 micromachines-14-00088-f013:**
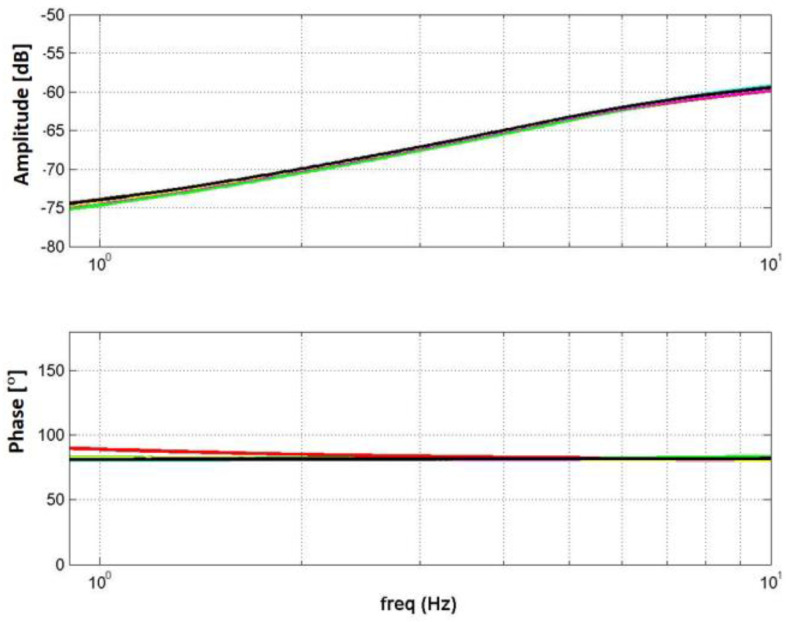
Bode plot of current as a function of input voltage amplitude and load.

**Figure 14 micromachines-14-00088-f014:**
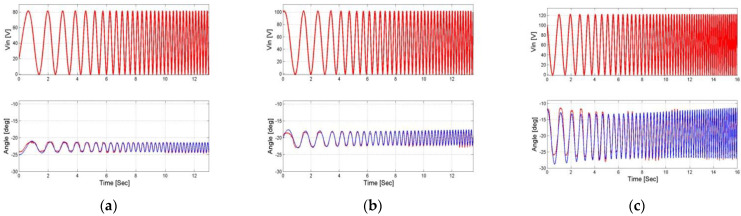
Comparison of uniform transfer function vs. measured output angle results when the first load is applied. (**a**) Vin = 80 V, (**b**) Vin = 100 V, and (**c**) Vin = 120 V.

**Figure 15 micromachines-14-00088-f015:**
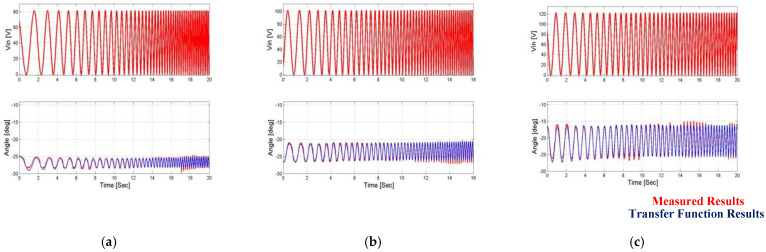
Comparison of uniform transfer function vs. measured output angle results when the second load is applied. (**a**) Vin = 80 V, (**b**) Vin = 100 V, and (**c**) Vin = 120 V.

**Figure 16 micromachines-14-00088-f016:**
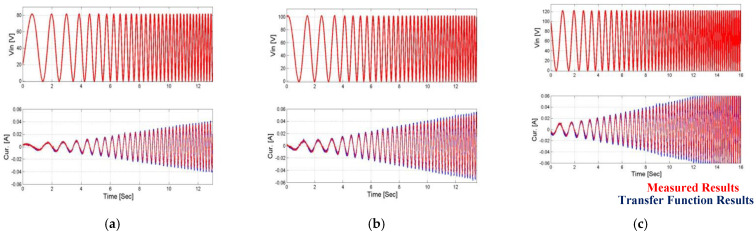
Comparison of uniform transfer function vs. measured output current results when the first load is applied. (**a**) Vin = 80 V, (**b**) Vin = 100 V, and (**c**) Vin = 120 V.

**Figure 17 micromachines-14-00088-f017:**
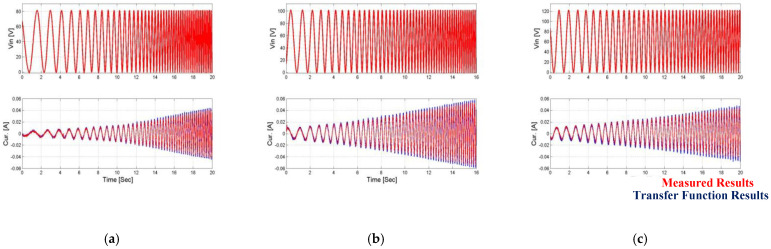
Comparison of uniform transfer function vs. measured output current results when the second load is applied. (**a**) Vin = 80 V, (**b**) Vin = 100 V, and (**c**) Vin = 120 V.

**Figure 18 micromachines-14-00088-f018:**
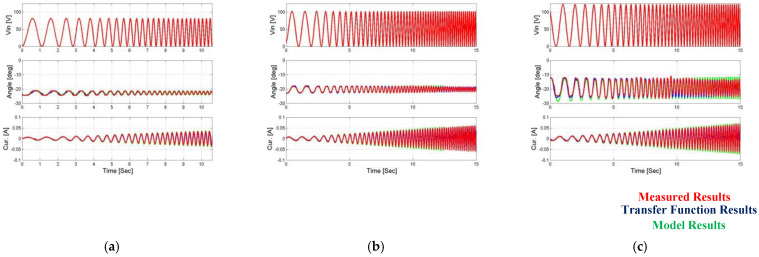
Comparison of uniform transfer function results’ model results vs. measured output results when the first load is applied. (**a**) Vin = 80 V, (**b**) Vin = 100 V, and (**c**) Vin = 120 V.

**Figure 19 micromachines-14-00088-f019:**
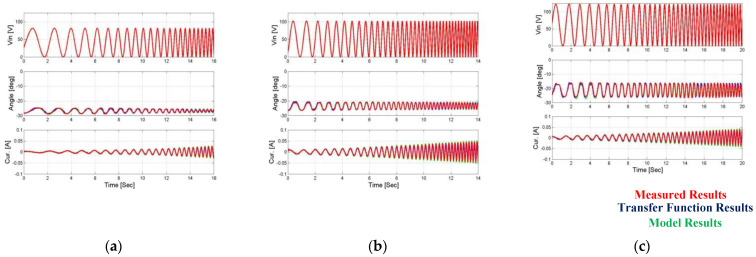
Comparison of uniform transfer function results’ model results vs. measured output results when the second load is applied. (**a**) Vin = 80 V, (**b**) Vin = 100 V, and (**c**) Vin = 120 V.

**Table 1 micromachines-14-00088-t001:** Model variables.

Name	Description	Units
*V_in_*	Voltage developed on the piezostack electrodes	[Volt]
*V_act_*	Total voltage in the PEM	[Volt]
*F_act_*	Total force in the PEM	[Newton]
*F_ext_*	External force	[Newton]
*v*	PEM velocity	[meter/sec]
*x*	PEM displacement	[meter]

**Table 2 micromachines-14-00088-t002:** Model parameters.

Name	Description	Units
*d_33_*	Piezoelectric coefficient	[meter/Volt] or [Coulomb/Newton]
*C_m_*	Mechanical stiffness	[Newton/meter]
*m*	Effective mass	[Kg]
*K_d_*	Viscous friction	[Newton·sec/meter]
*R*	Actuator internal ohmic resistance	[Ohm]
*C*	Actuator capacitance	[F]

**Table 3 micromachines-14-00088-t003:** Hysteresis equation parameters.

Descent Curve				Ascend Curve			
*P_dn1_*	1.7× 10^−6^	*B_dn1_*	−17.67 × 10^−3^	*P_up1_*	42.36 × 10^−6^	*B_up1_*	−23.55 × 10^−3^
*P_dn2_*	4.54 × 10^−4^	*B_dn2_*	2.1	*P_up2_*	3.65 × 10^−3^	*B_up2_*	2.41
*P_dn3_*	2.51 × 10^−2^	*B_dn3_*	0.53	*P_up3_*	−53 × 10^−3^	*B_up3_*	1.72
*P_dn4_*	0.438			*P_up4_*	0.137		
*P_dn5_*	124.94						

**Table 4 micromachines-14-00088-t004:** Amplification mechanism equations results.

Name	Description	Units
L_1_	Arm of the first lever	mm
L_2_	Arm of the second lever	mm
α_34_	Angular movement of the second stage	deg
α_out_	Angular movement of the output stage	deg

**Table 5 micromachines-14-00088-t005:** Amplification mechanism equations’ parameters.

Name	Description	Units
X_12_	Distance between points 1 and 2 in a horizontal direction.	60.43
Y_12_	Distance between points 1 and 2 in a vertical direction.	13
R_12_	Distance between points 1 and 2.	61.81
Rb_1_	Movement radius of the first lever.	2
X_13_	Distance between points 1 and 3 in a horizontal direction.	65.31
Y_13_	Distance between points 1 and 3 in a vertical direction.	20.50
Rb_2_	Movement radius of the second lever.	4.36
R_34_	Distance between points 3 and 4.	30
X_35_	Distance between points 3 and 5 in a horizontal direction.	36
Y_35_	Distance between points 3 and 5 in a vertical direction.	0.044
Rb_3_	Radius of movement of the third lever.	4.48
(In this table, all the units are mm)		

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
