# Peer review of "System Identification and Mathematical Modeling of A Piezoelectric Actuator through A Practical Three-Stage Mechanism"

_micromachines, 2022, doi:10.3390/mi14010088_

Round 1
Reviewer 1 Report
In this paper, a simple system identification algorithm is applied in a piezoelectric actuator, and the system level method of piezoelectric actuator is adopted, which provides a solid modeling background, domain knowledge and system identification method for further research in the field of piezoelectric actuators. The authors show that the verified simple system identification algorithm can be improved on the model and system identification method to be applicable to other PEMS or other amplification mechanism methods. This paper has a detailed evaluation of each module, which is good.
Please answer following comments:
1. For ‘4.2.2Position feedback Transfer Function Adjusting’ in this paper,when the input voltage is 100 V or 120 V and the 1st Load is applied, the fitting effect between the measured value of position Angle and the result of the transfer function is not obvious.
2. When the input voltage is 80 V 、100 V 、 120 V and the 2nd Load is applied, the fitting effect between the measured value of position Angle and the result of the transfer function is not obvious. It is suggested to systematically identify the constants in equation (13) again to find a more suitable one.
3. In ‘4.3Model verification’ of this paper, the author compares the results of the evaluation function of position and current feedback with the results of the whole system model and the actual measurement results. The effect shown in Figure 18 (c) is not obvious, that is, when the input voltage is 120V, the fitting effect of the position Angle is not obvious.
In general, I think this paper models each unit of the piezoelectric actuator system, and then combines these units, and considers their coupling at the same time, and uses a new simple algorithm to verify the system, which is of great importance to the research in the field of piezoelectric brakes. The paper is well written and easy to understand.
Reviewer 2 Report
A study on system identification and mathematical modeling of piezoelectric actuator in practical three-stage mechanism is presented in this paper. Even it is interesting, some irregular terminology and expression errors are presented in this paper, which need the authors to correct elaborately
1) The quality of Figure 1 should be improved for better reading comprehension effects, especially the amplification mechanism.
2) Some expression mistakes exist in the manuscript. Please correct them carefully and smooth the English description in the whole article.
3) It is difficult to understand the equations (8-11), and how to derive it, please clarify.
4) How to obtain the experimental results that used to conduct the system identification, and the test setup both should be exhibited.
5) All figures in this paper should be improved.
Reviewer 3 Report
Dear Authors,
The manuscript should be revised before it is published.
My review report is in the attachment.
Kind Regards

Round 2
Reviewer 2 Report
It can be accepted for publication with minor revision.
Reviewer 3 Report
Dear Authors,
I have no further questions and remarks. The manuscript can be published after minor revision.
Kind Regards